# Impact of Raised without Antibiotics Measures on Antimicrobial Resistance and Prevalence of Pathogens in Sow Barns

**DOI:** 10.3390/antibiotics11091221

**Published:** 2022-09-08

**Authors:** Alvin C. Alvarado, Samuel M. Chekabab, Bernardo Z. Predicala, Darren R. Korber

**Affiliations:** 1Chemical and Biological Engineering, University of Saskatchewan, 57 Campus Dr., Saskatoon, SK S7N 5A9, Canada; 2Prairie Swine Centre Inc., Box 21057, 2105–8th St. East, Saskatoon, SK S7H 5N9, Canada; 3Food and Bioproduct Sciences, University of Saskatchewan, 51 Campus Drive, Saskatoon, SK S7N 5A8, Canada

**Keywords:** raised without antibiotics, WGS metagenomics, antimicrobial resistance genes, gut microbiome, nasopharynx microbiome

## Abstract

The growing concern over the emergence of antimicrobial resistance (AMR) in animal production as a result of extensive and inappropriate antibiotic use has prompted many swine farmers to raise their animals without antibiotics (RWA). In this study, the impact of implementing an RWA production approach in sow barns on actual on-farm antibiotic use, the emergence of AMR, and the abundance of pathogens was investigated. Over a 13-month period, fecal and nasopharynx samples were collected at 3-month intervals from sows raised in RWA barns and sows in conventional barns using antibiotics in accordance with the new regulations (non-RWA). Whole genome sequencing (WGS) was used to determine the prevalence of AMR and the presence of pathogens in those samples. Records of all drug use from the 13-month longitudinal study indicated a significant reduction in antimicrobial usage in sows from RWA barns compared to conventional non-RWA barns. Antifolates were commonly administered to non-RWA sows, whereas β-lactams were widely used to treat sows in RWA barns. Metagenomic analyses demonstrated an increased abundance of pathogenic *Actinobacteria*, *Firmicutes*, and *Proteobacteria* in the nasopharynx microbiome of RWA sows relative to non-RWA sows. However, WGS analyses revealed that the nasal microbiome of sows raised under RWA production exhibited a significant increase in the frequency of resistance genes coding for β-lactams, MDR, and tetracycline.

## 1. Introduction

Antibiotic overuse and misuse in human and veterinary medicine as well as in agriculture has intensified the emergence and spread of antimicrobial-resistant diseases, making it one of the most serious risks to public health according to World Health Organization [1]. Globally, animal agriculture accounts for more than half of all antibiotic use, which was estimated to be about 131,000 tons in 2013 and is expected to exceed 200,000 tons by 2030 [2]. In Canada, 78% of antimicrobials sold in 2019 (~1.0 million kg) were intended for use in food-producing animals, with the majority (both in kilograms and adjusted for the number of animals and their weights (biomass)) were distributed for use in pig production [3]. This trend has been consistent worldwide [4,5,6,7,8,9]. Tetracyclines, macrolides, and penicillins (β-lactams) are among the top classes of antimicrobial sold to the Canadian swine sector [3].

Due to risks associated with antimicrobial resistance (AMR) as a result of increased antibiotic use, Canada has implemented more stringent regulations on the use of antibiotics in livestock production. Since December 2018, all medically important antimicrobials for veterinary use must be sold by prescription only, and the use of antibiotics in animal feed is prohibited. Additionally, a number of pig producers have shifted to raising pigs without using any antibiotics (RWA), motivated by the premium price paid by processors for pigs raised completely without antibiotics. However, it has been reported that as swine producers transitioned to antibiotic-free production, the microbial load in the production environment gradually increased, eventually leading to the animals succumbing to the increased microbial challenge, after which severe disease outbreaks began to occur. Thus, the impact of RWA production on the emergence of AMR and prevalence of pathogens in the barn was explored further in this study.

According to Statistics Canada, the Canadian hog industry had a total inventory of around 14.11 million hogs in January 2022, including 5.17 million piglets, 7.68 million grower–finishers, and 1.24 million sows and gilts [10]. Sows, or adult female pigs who have given birth (farrowed) at least once, play a critical role in swine production. Sows are among the oldest animals in a swine barn. They are first bred at the age of 24–30 weeks and have a gestation period of about 115 days. When sows are about to farrow, they are moved to a farrowing room where they stay for 3–4 weeks until they wean their litter, and then they are bred again. Broad-spectrum antibiotics are usually fed or given by injection at farrowing and for a few days afterwards, because sows are susceptible to a number of infections during this period [11]. Given that sows stay longer in the barn, they are more likely to have been exposed to most antibiotics as well as pathogens [12], making the sow stage significant to investigate the impacts of RWA.

Furthermore, the Canadian Integrated Program for Antimicrobial Resistance Surveillance (CIPARS), which monitors antimicrobial usage in the Canadian swine industry, is mostly focused on grower-finisher pigs, with no data on sows available [13]. The impact of antibiotic treatments on the prevalence of pathogens and antimicrobial-resistant genes (ARGs) in the gut microbiome of piglets and grower-finisher pigs has previously been examined [14,15,16,17]; however, little is known about the potential differential effects of these treatments on the abundance of pathogens and ARGs in the gut and nasal microbiome of sows [12]. Additionally, while the gut microbiome of pigs has been extensively studied, the impact of antimicrobial treatment on the nasal microbiome has yet to be investigated [18,19]. There is mounting evidence that the nasal microbiota regulates local immunity and contributes to swine respiratory health [19]. Hence, the main objectives of this study were to characterize the gut and nasopharynx microbiome of sows raised under two different production conditions (RWA vs. non-RWA), monitor their antimicrobial usage, and investigate the impact of RWA measures on the prevalence of AMR and the occurrence of pathogens found in the gut and nasopharynx of sows.

## 2. Results

### 2.1. Antifolates Are Largely Administered to Non-RWA Sows While β-Lactams Are Still Widely Used for Treatment of Sows in RWA Barns

Records of all drug use for animal treatments as well the actual use of antibiotics in the participating barns were collected regularly from November 2019 to November 2020. These records included the type of drug, dosage, the number and age of treated animals, the cause of treatment, location in the barn, and the date of drug administration. The information and amounts of drugs given were cross-referenced against the Provincial Veterinary Services and swine drug treatment databases (https://www.drugs.com/vet/swine-a.html, accessed on 9 July 2021). The antibiotics mostly given to sows during the monitoring period belonged to three classes: antifolates (Trimidox), β-lactams (Penicillin G, Ampicillin, Ceftiofur), and tetracyclines (Biomycin).

Table 1 shows a list of antibiotics that were administered in RWA and non-RWA barns along with their corresponding quantity in milligrams (mg) and DDDvetCA, which is the Canadian defined daily dose (average labeled dose) in milligrams per kilogram pig weight per day (mg drug/kg animal/day), in accordance with the Canadian Integrated Program for Antimicrobial Resistance Surveillance 2016 report [20]. The data on antibiotic type and dosages were normalized using the number of sows treated during the surveillance period in order to report the values as DDDvetCA. On average, a total of 553 g of antibiotics were administered to 78 non-RWA sows, corresponding to 1285 mg/day/kg cumulative DDDvetCA. However, RWA sows (*n* = 22) received a total of 130 g of antibiotics, corresponding to 204 mg/day/kg cumulative DDDvetCA. The estimated total antimicrobial usage for sows in RWA barns was reduced by nearly 4.3-fold compared to usage in non-RWA barns. The number of treated sows in RWA barns was 3.5-fold lower and received 6.3-fold less cumulative DDDvetCA value than the sows in non-RWA barns. Sows in non-RWA barns were treated with antifolates (77% of total antibiotics administered), β-lactams (1.4%), and tetracycline (22%), which respectively corresponded to 72%, 1.3%, and 27% of the cumulative DDDvetCA value. In contrast, RWA sows were treated with β-lactams (82%) and tetracycline (18%), which respectively corresponded to 85% and 15% of the cumulative DDDvetCA value (Table 1). Due to potential interpretation bias caused by the lack of consistent criteria for the different treatment categories/reasons across participating barns (e.g., one barn may call the reason for treatment ‘infection’, while others may list the same symptoms as ‘limping’ or ‘respiratory’), these data were not included in the analysis. Nevertheless, the six most prevalent reasons for treatment recorded during the 13-month monitoring period included infection, injury, limping, respiratory impairment, and symptoms associated with gestation such as difficulty farrowing, vaginal discharge, and mastitis.

### 2.2. RWA Sows Exhibit More Pathogenic Actinobacteria, Firmicutes, and Proteobacteria in Nasopharynx Samples

Metagenomic taxonomy profiling was performed on samples from sow feces and nasal swab samples collected from RWA and non-RWA barns. Bacterial sequences accounted for nearly 99% of all sequenced reads matching the k-mer markers. Analysis through the CosmosID bioinformatics platform resulted in the identification of 395 bacterial species belonging to 150 genera, 24 classes, and 13 phyla (data not shown). The sows’ fecal microbiomes were dominated by *Firmicutes* (65–68%), followed by *Actinobacteria* (9–13%), and *Bacteriodetes* (6–7%). The nasal swab samples, however, exhibited a fairly comparable relative abundance at the phylum level, including *Firmicutes* (39–54%) and *Proteobacteria* (19–31%).

The total species/strains present in the microbiome were further analyzed to determine the prevalence of pathogens (the pathome) which is represented by the subset of human and/or animal risk group (RG) 2 and RG3 organisms. From our analysis, 89 and 96 pathogenic strains were found in feces and nasal swabs samples, respectively (See Appendix A). The total prevalence of pathogens (sequence frequency) in the sow feces was comparable between the non-RWA and RWA barns with an overall reduction in the RWA samples (Figure 1). However, at the phylum level, sow feces in RWA barns exhibited a significant decrease in the frequency of pathogens over time, including microorganisms belonging to *Firmicutes* (Figure 2A, Appendix A). Alternatively, the total prevalence of pathogens in the nasal swab samples collected from sows was remarkably higher in RWA barns compared to non-RWA (Figure 1). At the phylum level (Figure 2B), the pathogenic *Actinobacteria*, *Firmicutes*, and *Proteobacteria* were significantly more frequent in the nasal swab samples collected from RWA sows (Appendix A).

### 2.3. RWA Practices Increased AMR in the Sow Nasopharynx but Not in the Sow Gut

To determine the effect of the RWA production approach on the prevalence of ARGs over time in both the gut and nasopharynx microbiomes of sows, the resistome profiles from the WGS data were compared and yielded the frequency of ARGs present in the samples based on absolute number of reads. As shown in Figure 3, the ARGs were found to belong to six main classes: aminoglycoside (11–32%), β-lactam (2–26%), macrolide (16–32%), phenicol (0.04–0.88%), multi-drug resistance (MDR; 3–13%), and tetracycline (22–46%). Over 13 months of surveillance monitoring, the ARG frequency in sow feces in RWA barns was not significantly different (*p* = 0.072) from non-RWA barns (Figure 3 and Figure 4A). However, the ARG frequency in samples collected from sow nasopharynx was significantly higher (*p* = 0.016) in RWA barns compared to non-RWA barns (Figure 3). Relative to sows raised under non-RWA conditions, the frequency of ARGs in the nasal swab samples from sows in RWA barns significantly increased for β-lactam (*p* = 0.004), MDR (*p* = 0.03), and tetracycline (*p < 0.002*) drug classes (Figure 4B).

Figure 5 shows the frequencies of the ten most abundant ARGs in sow feces and nasal swab samples from RWA and non-RWA barns. Through comparative gene frequency (heat map) analysis of ARG clusters, tetracycline resistance genes *tetW* and *tetQ* were the most frequently found in fecal samples, and the β-lactam resistance gene *blaROB1* was the most frequently found in nasal swab samples, followed by Aminoglycoside (*ant9 la*) and then by MDR (*lsaE*) (Figure 5A). Furthermore, PCoA ordination of ARGs significantly clustered the effect of RWA on the resistome from sow nasal swab samples (*p* = 0.016), but not from sow feces (*p* = 0.072) (Figure 5B).

## 3. Discussion

The impact of antibiotic treatments on the prevalence of pathogens and ARGs in the gut microbiome of piglets and grower-finisher pigs has previously been examined [14,15,16,17]; however, limited information is available on the potential differential effects of these treatments on the abundance of pathogens and ARGs in the gut and nasal microbiome of sows. The monitoring of antimicrobial usage in the Canadian swine industry through the Canadian Integrated Program for Antimicrobial Resistance Surveillance (CIPARS) is mainly focused on grower-finisher pigs, with no data on sows available [13]. Similarly, most reports of antimicrobial usage in the United States are mostly focused on nursery and grower-finisher pigs, and data on sows are not well-documented [21]. In Europe, a study conducted by Echtermann et al. [22] involved the monitoring of antimicrobial usage from 71 Swiss non-RWA farrow-to-finish farms in 2017. Based on antimicrobial data expressed in terms of the number of defined daily doses Switzerland (nDDDch) in milligrams per kilogram pig weight, the most prevalent treatment was penicillin (β-lactams), which accounted for 57% of the nDDDch, while tetracyclines were minimal (2%), and antifolates were not reported [22]. These findings contrasted with non-RWA practices in Canadian operations shown in our study, where sows were treated with more antifolates (72% of the cumulative DDDvetCA value) and less β-lactams (1.3%), but consistent with RWA practices (85% β-lactams and no antifolates). These differences could be attributed to a number of factors such as geographic location, animal management, and age of pigs, among others. In non-RWA sows, antifolates were used to treat respiratory impairment and limping symptoms, while β-lactams were administered to sows in participating RWA barns for symptoms related to gestation, as well as infection and injuries (visible swellings). Antifolates are widely used in Canadian swine production; these drugs are antibacterial, immunomodulatory, and chemotherapeutic agents but the mechanism of resistance is poorly documented and is likely linked to inducing efflux multidrug-resistance [23,24]. On the other hand, resistance to β-lactams and tetracyclines have been proven to be globally ubiquitous in pig microbiomes yet still remain in use and clinically valuable [25,26,27].

In order to assess the impact of the RWA approach on the prevalence of pathogens and ARGs, fecal and nasal swab samples were collected from sows in RWA and non-RWA barns. Analyses of the metagenomes showed increased abundance of pathogenic *Actinobacteria*, *Firmicutes*, and *Proteobacteria* in the nasopharynx microbiome of sows raised under the RWA program relative to non-RWA sows. This could be attributed to the reduced antimicrobial usage of sows in RWA barns, which was about 4.3-fold lower than in non-RWA barns. However, in other studies the impact of antibiotics administration on the nasal microbiota of pigs was variable. In a study by Correa-Fiz et al. [18], the non-RWA practice at early stage of life caused significant increase in the relative abundance of *Campylobacter*, but exhibited a lower relative abundance of other potentially pathogenic bacteria in the nasal microbiota of piglets. Similarly, the administration of antibiotics caused considerable variation in the relative abundance of some pathogenic bacteria in the nasal microbiota of pigs; the variation was dependent on dosing regimen [28], age of pigs [18] and type of antimicrobial treatment [29]. In this study, the decrease in the relative abundance of *Actinobacteria* over time in the nasal microbiota of sows raised under the RWA program could be associated with the stage of growth. In a study by Slifierz et al. [30], only the relative abundance of *Actinobacteria* decreased over time among the bacterial phyla in the nasal microbiota of pigs, and the authors suggested that this temporal shift may have been caused by aging. This decreasing trend, however, was not evident in the nasopharynx samples from non-RWA sows because of their extensive exposure to antibiotic treatments which can disturb the nasal microbiome of pigs [28].

On the other hand, WGS data from the sow gut demonstrated a higher prevalence of pathogenic *Firmicutes* in non-RWA samples than in RWA samples. This is consistent with the findings of Sun et al. [31], where an increase in the abundance of *Firmicutes* in sow fecal samples was observed after the administration of antibiotics. *Firmicutes* was previously found to be the most abundant phylum in the core and variable microbiome of acute diarrheal patients (dysbiosis) and/or a source of ARGs in the infected gut of individuals treated with antibiotics [32]. In addition, the relative abundance of pathogenic bacteria, such as *Clostridium* and *Corynebacterium*, increased in the gut microbiota of finishing pigs administered with lincomycin [15]. This increase, according to the authors, could be due to their antibiotic-resistant properties [15].

In this study, there was no significant difference in the frequency of ARGs detected in the sow feces from RWA and conventional (non-RWA) barns. However, the metagenomics results from the nasal microbiome of sows raised under RWA production exhibited a significant increase in the frequency of resistance genes coding for β-lactams, MDR, and tetracycline compared to non-RWA sows. This could be associated with sows being exposed to higher concentrations of β-lactams and tetracycline in RWA barns relative to non-RWA barns. On the other hand, the impact of the increased usage of these types of antibiotics in RWA barns was not evident in the fecal microbiome of sows, which suggests that the ARGs in the nasal microbiome could be more susceptible to antibiotic exposure than those in the fecal microbiome. In the study by Holman et al. [33] on the impact of antibiotic treatments on the fecal and nasal microbiota of feedlot cattle, antibiotic resistance determinants in the nasal microbiome were more significantly affected by antibiotic treatment than those in the fecal microbiome. To date, information on the correlation between antimicrobial usage and frequency of ARGs in the nasal microbiome of sows is very limited. However, several studies have shown the emergence of a substantial number of antimicrobial resistance determinants in the swine gut microbiome, even in the absence of antimicrobial exposure [16,17,34,35,36]. For instance, in a study by Looft et al. [35], numerous tetracyclines genes, including *tetB(P)* and *tetQ,* were frequently found in fecal samples from piglets with no antibiotic exposure. Similarly, our present study demonstrated a higher abundance of tetracycline resistance genes *tetW* and *tetQ* in sow fecal samples. The emergence of antimicrobial resistance determinants in swine facilities without direct antimicrobial exposure could be related to the inclusion of high doses of heavy metals such as zinc and copper in swine diets [36,37,38]. Cross-resistance, as well as co-selection and co-resistance between antimicrobial resistance determinants and heavy metal resistance genes, could potentially explain why antimicrobial resistance genes persist in the pig gut even without prior antimicrobial administration [39,40].

## 4. Materials and Methods

### 4.1. Experimental Design and Sample Collection

A 13-month longitudinal study of swine barns that adopted the RWA program, and conventional barns using antibiotics in line with the new regulations (non-RWA), was conducted by collecting fecal and nasopharynx samples from sows at 3-month intervals; these samples were then subjected to whole genome sequencing (WGS) and compared using bioinformatics. During the 13-month period, regular collection of metadata was also conducted, and was comprised of all records of administered antibiotic drugs and illnesses or treatment reasons from the two types (RWA or non-RWA) of participating barns. Table 2 summarizes the type, name, dosages, and routes of administration of antibiotics to sows in RWA and non-RWA barns. Sows in the participating non-RWA barns were fed with wheat/barley-based diets while those in RWA barns had wheat-based and corn-based diets.

A total of five regular animal sampling time points was conducted over the monitoring period. At each time point, fresh fecal samples were collected aseptically from three third-parity sows (approximately 18–22 months old) and stored in sterile 50 mL tubes. The nasal swabs were collected from the same animals by following similar procedures described in the Swab Collection and DNA Preservation System (Norgen Biotek Corp., Thorold, Canada). The nasal swab samples were analyzed to detect potential subsets of respiratory viruses along with other microorganism categories, and their associated antimicrobial resistance genes (ARGs), that may not be well represented in the fecal samples. All samples were stored at 4 °C in a styrofoam container and shipped within 24 h of collection to the laboratory for storage at −80 °C and subsequent analyses. The handling of samples followed the guidelines outlined in the CDC’s Biosafety in Microbiological and Biomedical Laboratories (BMBL) manual for Level 1 biologic materials [41].

### 4.2. Whole Genome Sequencing (WGS) and Sequence Analyses

The total complement of ARGs (the resistome), bacterial diversity, as well as the prevalence of pathogens in the collected samples were identified by random shotgun next-generation sequencing (NGS) using an Illumina HiSeq platform (Omega-Bioservices, Norcross, GA, USA). Sample handling was performed in accordance with the sequencing service procedures and then shipped to Omega-Bioservices for DNA extraction, data quality determination, and NGS. A detailed workflow method of a health metadata-based management approach to compare and quantify WGS data targeting the occurrence of antimicrobial resistance and pathogens in Canadian swine barns was reported previously by the team [40]. DNA extraction from 1 g of sample material, and validation of the purity and yield of the DNA, were carried out using the Mag-Bind Universal Pathogen DNA Kit (Omega Biotek, Inc. Norcross, GA, USA) and Quant-iT™ PicoGreen™ ds DNA System kit (ThermoFisher Scientific, Pittsburgh, PA, USA), respectively. Shotgun NGS libraries were made from DNA using Kapa Biosystems Prep Kit according to manufacturer’s instructions (Roche^®^, KK2103 Pleasanton, CA, USA). Samples corresponding to distinct collection points were run on one lane of a HiSeq4000/X Ten instrument (Illumina, San Diego, CA, USA), generating a total of 100–120 GB of 150-bp paired-end data reads. Each sample generated two FASTQ files (R1 Forward read and R2 Reverse read), producing an average minimum of ~30 million reads (MReads) that were shared through the Illumina BaseSpace Sequence Hub. The sequences were then subjected to quality control (denoising and adaptor trimming) and reported using the MultiQC tool (v1.11 SciLifeLab, Stockholm, Sweden) at https://multiqc.info/, accessed on 9 July 2021 before being uploaded to the platform for metagenomic analysis (CosmosID Inc., Rockville, MD, USA).

### 4.3. Prevalence of Pathogens and Resistome

A subset of the taxonomic profiles belonging to bacterial pathogens was used to identify the total complement of pathogens (the pathome). As described in a study by Chekabab et al. [14], microbes were classified and identified on the species-, subspecies-, and strain-levels. The relative abundance and frequency of the identified organisms were quantified using GenBook comparators and the GENIUS software implemented within the CosmosID algorithm. This taxonomic profiling included a subset of pathogenic bacteria (the pathome), which was manually assigned to human and animal risk groups (RG2, or RG3).

The ARGs in the microbiome (resistome) were identified and quantified by comparing unassembled sequence reads to CosmosID’s curated ARG database. NCBI- RefSeq, PATRIC, M5NR, ENA, DDBJ, CARD, ResFinder, ARDB, and ARG-ANNOT were among the inputs utilized by the ARG database in the platform. These databases contain nearly 4000 distinct ARGs based on percent gene coverage as a function of the gene-specific read frequency in each sample. Using a classification system derived from two antimicrobial resistance pipelines (https://megares.meglab.org/, accessed on 9 July 2021 and https://card.mcmaster.ca/, accessed on 9 July 2021 the resulting ARG profile table was then clustered into 16 drug resistance classes and 7 resistance mechanisms [42].

### 4.4. Statistical Analysis

Frequency tables with taxa and ARG were subjected to univariate and multivariate analyses for diversity, ordination, and differential frequency. The species richness (Shannon alpha diversity indices and beta diversity distance matrices) was calculated. Two data sets from each type of farm (non-RWA and RWA) were included as biological replicates for statistical analysis.

Principal Coordinate Analysis (PCoA) was performed to cluster the readout frequencies of pathogen species and ARG class in the samples (Bray-Curtis distance matrix; community structure). PERMANOVA analysis was used to identify significantly different readouts. Both PCoA and PERMANOVA analyses were conducted using the CosmosID platform. Individual ARG classes of drug resistance were compared using two-way parametric ANOVA (GraphPad Prism v7.00, San Diego, CA, USA) with non-RWA and RWA as barn groups, and fecal, nasal and time-point repeated measurements as sub-groups.

## 5. Conclusions

The adoption of RWA measures in sow barns to reduce the total on-farm usage of antibiotics, and consequently to mitigate the emergence of AMR, has caused significant shifts in the diversity and abundance of certain pathogens and ARGs in the gut and nasopharynx microbiome of sows. During the 13-month monitoring period, whole genome sequence analyses revealed that sows raised under the RWA program had a higher frequency of pathogens in the nasopharynx, and a lower frequency of pathogens in the gut relative to sows in conventional non-RWA barns. On the other hand, the reduction of antibiotic usage in RWA barns resulted in an increased abundance of ARGs in sow nasopharynx but had no significant impact on ARG frequency in sow feces. An expanded longitudinal monitoring with more participating non-RWA and RWA barns over a longer timeframe is needed to more definitively validate the correlations and trends observed in this study regarding the impact of reduced antibiotic exposure on the frequency of ARGs and prevalence of pathogens in the gut and nasopharynx microbiome of older animals like sows.

## Figures and Tables

**Figure 1 antibiotics-11-01221-f001:**
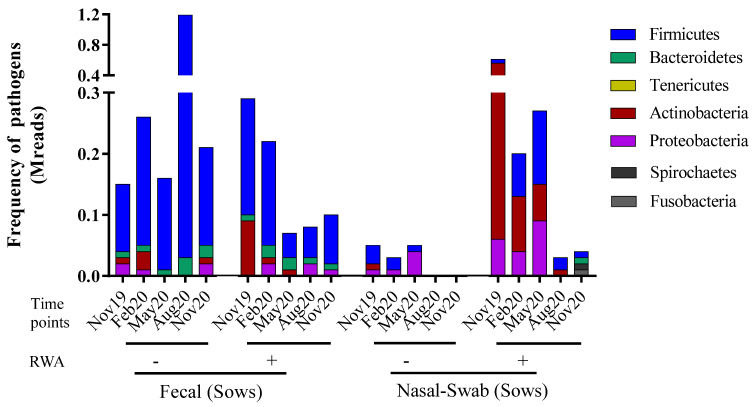
Metagenomic taxonomy profiling at the phylum level from sow feces and nasal swab samples collected from non-RWA − (minus sign) and RWA + (plus sign) barns. The stacked bars represent the average frequency of the major bacteriomes from each type of barn.

**Figure 2 antibiotics-11-01221-f002:**
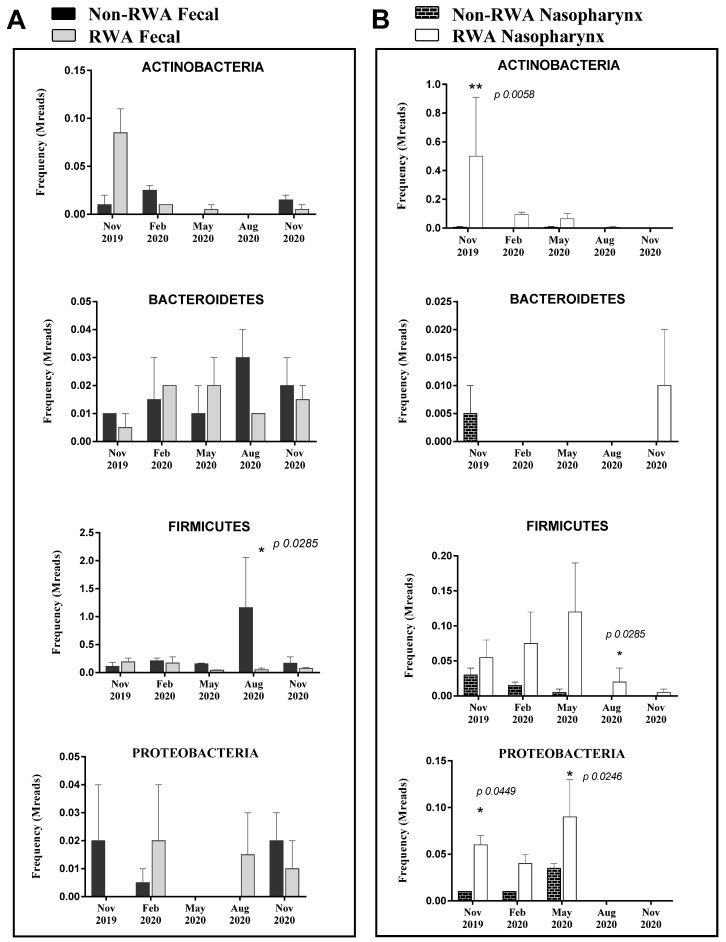
Pathogen prevalence in sow feces (**A**) and sow nasopharynx (**B**) samples. The bars represent the averaged frequency of the pathogens obtained from all bacteriome phyla by extracting the subset of human and/or animal Risk Group 2 species. Two data sets from each type of farm (non-RWA and RWA) were included as biological replicates for statistical analysis. ANOVA 2-way analysis with repeated measures comparing RWA vs. non-RWA with Bonferroni’s correction; * *p* < 0.05, ** *p* < 0.01.

**Figure 3 antibiotics-11-01221-f003:**
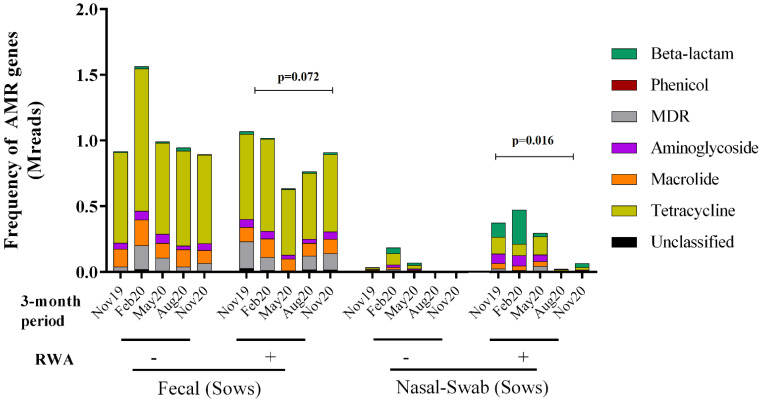
Metagenomic resistome showing the frequency of antibiotic resistance genes (ARGs) clustered into six classes: tetracycline, aminoglycoside, macrolide, phenicol, β-lactam, and multi-drug resistance (MDR) and collected from sow feces and nasal swab samples in non-RWA − (minus sign) and RWA + (plus sign) barns. Two data sets from each type of barn (non-RWA and RWA) were included as biological replicates for PERMANOVA analysis of the non-RWA vs. RWA resistome profiles. The ARG frequency in sow feces in RWA barns was not significantly different (*p* = 0.072) from non-RWA barns (Figure 3 and Figure 4A). The ARG frequency in samples collected from sow nasopharynx was significantly higher (*p* = 0.016) in RWA barns compared to non-RWA barns.

**Figure 4 antibiotics-11-01221-f004:**
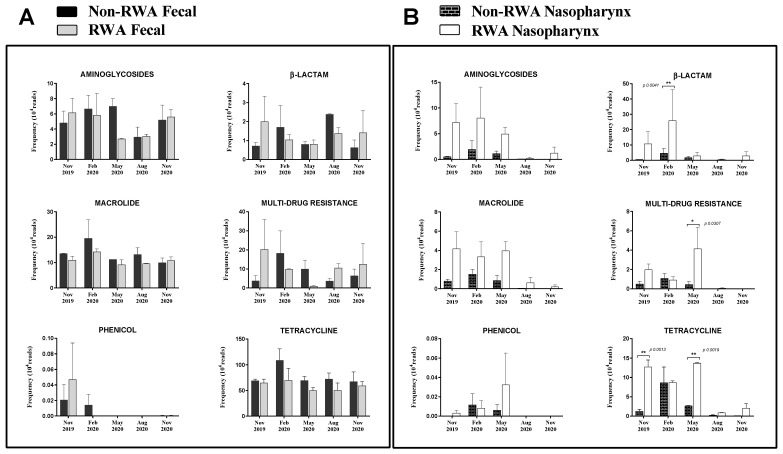
Resistomes from sow feces (**A**) and sow nasopharynx (**B**) in RWA + (plus sign) and non-RWA − (minus sign) barns. The frequency of antibiotic resistance genes (ARG) clustered in six classes: aminoglycoside, β-lactam, macrolide, MDR, phenicol, and tetracycline. Two data sets from each type of barn (non-RWA and RWA) were included as biological replicates for statistical analysis. ANOVA 2-way analysis with repeated measures comparing RWA vs. non-RWA with Bonferroni’s correction; * *p* < 0.05, ** *p* < 0.01.

**Figure 5 antibiotics-11-01221-f005:**
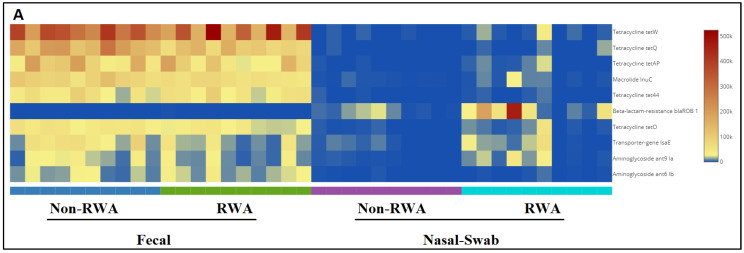
Gene frequency heat map comparative analysis of sow feces and nasal swabs from non-RWA vs. RWA barns, showing the frequencies of the 10 most abundant ARGs (**A**). PCoA and ordination PERMANOVA analysis of the RWA vs. non-RWA resistome profiles from sow feces and nasal swab samples (**B**).

**Table 1 antibiotics-11-01221-t001:** Quantification of antibiotics administered to sows in the participating barns.

Numbers of Animals	Type of Antibiotics	Antibiotics Used			
		Absolute Quantity		Relative Amount	
		mg	Percent	DDDvetCA ^1^	Percent
Non-RWA					
Sow (*n* = 78)	Antifolate	425,496	77%	925	72%
	Β-lactam	8000	1%	13	1%
	Tetracycline	119,500	22%	347	27%
	Total antibiotics (mg)	552,996		1285	
RWA					
Sow (*n* = 22)	β-lactam	107,085	82%	173	85%
	Tetracycline	23,250	18%	31	15%
	Total antibiotics (mg)	130,335		204	

^1^ DDDvetCA is the Canadian defined daily dose (average labeled dose) in milligrams per kilogram pig weight per day (mg drug/kg animal/day).

**Table 2 antibiotics-11-01221-t002:** Type, drug name, dosages, and route of administration of antibiotics to sows in non-RWA and RWA barns.

Barns	Type of Antibiotics	Drug Name	Dosage	Treatment Route ^1^
Non-RWA				
	Antifolate	Trimidox (trimethoprim & sulfadoxine)	1 mL/15 kg/day	IV or IM injection
	Tetracycline	Biomycin (Oxytetracycline)	1 mL/10 kg/day	IM or subcutaneous
	β-lactam	Penicillin G	6000 IU per kg (1 mL/50 kg)	IM injection
		Polyflex (ampicillin)	6 mg/kg/day	IM injection
		Excenel (ceftiofur)	3.0 mg/kg/day for 3 days	IM injection
RWA				
	β-lactam	Polyflex (ampicillin)	6 mg/kg/day	IM injection
		Penicillin G	6000 IU per kg (1 mL/50 kg)	IM injection
	Tetracycline	Biomycin (Oxytetracycline)	1 mL/10 kg/day	IM or subcutaneous

^1^ IM is intramuscular; IV is intravenous.

## Data Availability

DNA metagenomics sequencing data are available in Sequence Read Archive (SRA) (https://submit.ncbi.nlm.nih.gov/subs/sra/) with the accessions number PRJNA844237.

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
