# Peer review of "Impact of Raised without Antibiotics Measures on Antimicrobial Resistance and Prevalence of Pathogens in Sow Barns"

_antibiotics, 2022, doi:10.3390/antibiotics11091221_

Round 1
Reviewer 1 Report
In this well written manuscript, the authors have investigated the emergence of antimicrobial resistance and abundance of pathogens in sows raised without antibiotics as compared to sows raised using antibiotics.
Overall the authors have conducted a detailed study and presented convincing data that supports their conclusions. Following are a few minor comments that the authors could consider addressing in order to strengthen the manuscript:
· Please include sample size and significance testing information in the Figure legends where applicable
· Figure 2b shows that the Actinobacteria levels are significantly higher in the RWA nasopharynx sample of November 2019 as compared to RWA nasopharynx samples from subsequent months. Can the authors address the reason for this higher level of Actinobacteria levels in the first month?
Author Response
- Please include sample size and significance testing information in the Figure legends where applicable:
Response to comment: Thanks for noting this. We’ve added this information in the relevant figure legends to ensure this is clear:
- Sample size for collection of antibiotic treatments data: Non-RWA Sow (N=78), RWA Sow (N=22). See Table 1.
- Sample size for pathome and resistome profiling: A statement indicating “Two data sets from each type of farm (non-RWA and RWA) were included as biological replicates for statistical analysis.” has been added to legend captions for Figures 2-4.
- Significance testing: PermANOVA analysis of the non-RWA vs. RWA resistome profiles was conducted (see modified Fig 3 legend).
Relevant text in legend now reads “Two data sets from each type of barn (non-RWA and RWA) were included as biological replicates for permANOVA analysis of the non-RWA vs. RWA resistome profiles. The ARG frequency in sow feces in RWA barns was not significantly different (p=0.072) from non-RWA barns (Figures 3 and 4A). The ARG frequency in samples collected from sow nasopharynx was significantly higher (p=0.016) in RWA barns compared to non-RWA barns.”.
- Figure 2b shows that the Actinobacteria levels are significantly higher in the RWA nasopharynx sample of November 2019 as compared to RWA nasopharynx samples from subsequent months. Can the authors address the reason for this higher level of Actinobacteria levels in the first month?
Response to comment: Aging could be a factor in the decrease of the relative abundance of Actinobacteria over time in the nasal microbiota of RWA sows. In the study by Slifierz et al. (2015), only the relative abundance of Actinobacteria decreased over time among the bacterial phyla in the nasal microbiota of pigs and they suggested that this temporal shift may have caused by aging. This decreasing trend, however was not evident in the nasopharynx samples from non-RWA sows because of their extensive exposure to antibiotic treatments which can disturb the nasal microbiome of pigs (Mou et al. 2019).
See lines 253-259, where related text now reads “In this study, the decrease of the relative abundance of Actinobacteria over time in the nasal microbiota of sows raised under the RWA program could be associated with stage of growth. In a study by Slifierz et al. [30], only the relative abundance of Actinobacteria decreased over time among the bacterial phyla in the nasal microbiota of pigs and they suggested that this temporal shift may have caused by aging. This decreasing trend, however was not evident in the nasopharynx samples from non-RWA sows because of their extensive exposure to antibiotic treatments which can disturb the nasal microbiome of pigs [28].”.
Reviewer 2 Report
Antibiotic resistance genes (ARGs) have attracted extensive attention as an emerging environmental contaminant potentially threatening humans. The overuse of antibiotics in veterinary and human medicine promotes antibiotic resistance. In this manuscript, the authors have compared the antimicrobial usage in RWA and non-RWA barns and evaluated the impact of antibiotics exposure on pathogens diversity and ARGs in the gut and nasopharynx microbiome of sows. The manuscript presents an interesting research work. Issues that need to be addressed to improve the paper are as follows:
1. Please give more specific information on the age of pigs, animal management, administration regimen (dosages, routes of administration, time intervals), and type of feeds in the materials and methods. All the above factors might influence the results.
2. The correlations between antimicrobial exposure and ARGs are not obvious when considered carefully. In this study, there was no significant difference in the ARG frequency in sow feces between RWA barns and non-RWA barns. However, the ARG frequency in nasopharynx samples collected from sows was significantly higher in RWA barns compared to non-RWA barns. Please explain this discrepancy.
Author Response
Reviewer #2
- Please give more specific information on the age of pigs, animal management, administration regimen (dosages, routes of administration, time intervals), and type of feeds in the materials and methods. All the above factors might influence the results.
Response to comment: Samples were collected from third-parity sows (approx. 18-22 months old). Table 2 summarizes the type, name, dosages and routes of administration of antibiotics to sows in RWA and non-RWA barns. Sows in the participating non-RWA barns were fed with wheat/barley-based diets while the RWA barns had wheat-based and corn-based diets. See lines 315-318, 321, and 337-338.
- The correlations between antimicrobial exposure and ARGs are not obvious when considered carefully. In this study, there was no significant difference in the ARG frequency in sow feces between RWA barns and non-RWA barns. However, the ARG frequency in nasopharynx samples collected from sows was significantly higher in RWA barns compared to non-RWA barns. Please explain this discrepancy.
Response to comment: The significant increase in the frequency of resistance genes coding for β-lactams, MDR and tetracycline in the nasal microbiome of sows raised under RWA production could be associated with the exposure of sows to higher levels of β-lactams and tetracycline in RWA barns relative to non-RWA barns. However, the impact of the increased usage of these types of antibiotics in RWA barns was not evident in the fecal microbiome of sows, which suggests that the ARGs in the nasal microbiome could be more susceptible to antibiotic exposure than those in the fecal microbiome. In the study by Holman et al. (2019) on the impact of antibiotic treatments on the fecal and nasal microbiota of feedlot cattle, antibiotic resistance determinants in the nasal microbiome were more significantly affected by antibiotic treatment than those in the fecal microbiome. See lines 287-293.
Reviewer 3 Report
Interesting article. To facilitate reading, I suggest a table with an explanation of the acronyms. Also in line 138 there is a typo (RG)2 and RG3Author Response
Reviewer #3
- To facilitate reading, I suggest a table with an explanation of the acronyms.
Response to comment: We disagree on this since list of abbreviations aren’t common for a research article (vs a review) and the fact that we do/did provide definitions for all abbreviations, in the text, at first use.
- line 138 there is a typo (RG)2 and RG3
Response to comment: It is not a typo.
Round 2
Reviewer 2 Report
The authors have revised manuscript according to the review comments